# Aminotransferase Class I and II Gene Family in the Jinjiang Oyster (*Crassostrea ariakensis*): Genomewide Identification, Phylogenetic Analysis and Expression Profiles after Salinity Stress

Guangyu Ge [1,2], Zhihong Liu [1,2,3], Tao Yu [4], Liqing Zhou [1,2,3], Xiujun Sun [1,2,3] , Zhuanzhuan Li [1,2,3] and Yanxin Zheng [4,*]

1 State Key Laboratory of Mariculture Biobreeding and Sustainable Goods, Yellow Sea Fisheries Research Institute, Chinese Academy of Fishery Sciences, Qingdao 266071, China; gegy001@163.com (G.G.); liuzh@ysfri.ac.cn (Z.L.); zhoulq@ysfri.ac.cn (L.Z.); xjsun@ysfri.ac.cn (X.S.); zhuanzhuanlee@163.com (Z.L.)

2 Key Laboratory of Sustainable Development of Marine Fisheries, Ministry of Agriculture and Rural Affairs, Yellow Sea Fisheries Research Institute, Chinese Academy of Fishery Sciences, Qingdao 266071, China

3 Laboratory for Marine Fisheries Science and Food Production Processes, Laoshan Laboratory, Qingdao 266237, China

4 Changdao Enhancement and Experiment Station, Chinese Academy of Fishery Sciences, Yantai 265800, China; cdyutao@126.com

* Correspondence: zhengyanxin1989@163.com

**Abstract:** Aminotransferases are enzymes found in living organisms that catalyze transfer reactions between amino acids and keto acids, crucial for amino acid metabolism and synthesis. Aminotransferase classes I and II play a vital role in regulating osmolarity, protecting cells, and improving metabolic homeostasis and cellular fitness. To investigate the characteristics of the aminotransferase class I and II gene family and their roles in osmotic pressure regulation in the Jinjiang oyster (*Crassostrea ariakensis*), the gene structure, chromosomal localization, and phylogeny were characterized and the genes' expression in the gill under high-salt stress was analyzed. In this study, eighteen Aminotransferase class I and II genes, including *SPTLCa* and *SPTLCb*, *SPTa* and *SPTb*, *ALAT2a* and *ALAT2b*, *KAT3a* and *KAT3b*, and *ASTa*, *ASTb* and *ASTc*, were identified. The physicochemical properties of 11 family members were stable, with their instability factors less than 40. Subcellular localization prediction showed that aminotransferase classes I and II were localized in the cytoplasm or mitochondria. Chromosomal localization results showed that the 18 aminotransferase class I and II genes were located on eight chromosomes. All members of this gene family had the *Aminotran_1_2* structural domain which is associated with osmotic pressure regulation by adjusting the conversion reaction between amino acids and keto acids. Most gene expressions showed an initial increase followed by a decrease from 0 h to 12 h when the oysters were challenged by acute stress using artificial seawater with a salinity of 40. However, the expression of *CarAGT2* and *CarKAT3b* genes showed an increased trend with increasing stress time. This study systematically investigated the bioinformatics characteristics of the aminotransferase class I and II gene family in *C. ariakensis* and their role in osmotic pressure regulation, which provides scientific data for understanding the potential functions of these genes in physiological adaptation, thereby expanding the research on osmoregulation in bivalves.

**Keywords:** *Crassostrea ariakensis*; aminotransferase class I and II gene family; high-salt stress; expression pattern

**Key Contribution:** This study has identified a total of 18 members belonging to the aminotransferase class I and II gene family in *Crassostrea ariakensis*. All of these members possess the *Aminotran_1_2* structural domain which is associated with osmotic pressure regulation. These members exhibited significant variations in relative expression levels during acute high-salinity stress. This research contributes to a deeper understanding of the aminotransferase class I and II gene family in bivalves,

providing foundational data for further exploration into the mechanisms by which this family is involved in regulating responses to osmotic stress.

## 1. Introduction

Salinity is one of the most important environmental factors that affect the survival and growth of shellfish. Unlike the relatively stable deep sea, the salinity in intertidal and estuarine areas is highly unstable [1]. High temperatures, seasonal heavy rainfall, and human activities can cause fluctuations in salinity and affect the survival of aquatic organisms [2]. Shellfish, in particular, have to actively acclimatize to fluctuating changes in salinity because of their relatively fixed mobility [3]. Despite the evolution of regulatory mechanisms in bivalves to cope with salinity changes, a large number of deaths are still experienced when they were exposed to acute changes in salinity [4]. To date, the majority of research on the effects of salinity on osmoregulation has been focused on marine organisms, such as *Sinonovacula constricta*, *Crassostrea hongkongensis*, *Litopenaeus vannamei*, *Meretrix petechialis*, and *Tegillarca granosa* [5–8]. Those studies indicated that they could adapt to fluctuations in seawater salinity through their own regulatory mechanisms [9]. Bivalves necessitate continuous evolution to adapt to the ever-changing salinity environment and safeguard against potential harm arising from significant fluctuations in internal osmotic pressure. Among these bivalve mollusks, the regulation of both internal and external osmotic pressure is achieved through the utilization of free amino acids. A previous study showed that the content of free amino acids significantly increased in *C. angulate* and *C. virginica* after they were treated with different salinities [10]. In *C. gigas*, free amino acids first increased and then decreased under low-salt treatment, whereas they increased under high-salt treatment [11]. In addition, in order to maintain the osmotic pressure balance in the body, bivalves use organic substances such as aspartic acid, glutamic acid, and taurine to increase the osmotic pressure level in their body [12]. Aspartate aminotransferase, an important member of the aminotransferase class I and II family, is a key enzyme that catalyzes the generation of aspartic acid. Therefore, studying the role of the aminotransferase class I and II gene family in osmotic pressure regulation in *C. ariakensis* is of great significance.

Aminotransferases play a pivotal role as a crucial category of enzymes that is primarily responsible for the metabolism and transfer of amino acids. Aminotransferases are classified into five classes, including aminotransferase class I and II, aminotransferase class III, aminotransferase class IV, and aminotransferase class V. Aminotransferase classes I and II include aspartate aminotransferase, aromatic amino acid aminotransferase, alanine aminotransferase, and histidine phosphate aminotransferase [13]. The transfer of the amino group of the donor to the ketone group of the acceptor substrate to produce the corresponding α-amino acid is catalyzed by this gene family, in which serine palmitoyltransferase 2 (*SPTLC2*) encodes a subunit of serine palmitoyltransferase (*SPT*) which is the rate-limiting enzyme in sphingolipid synthesis [14]. In fish, alanine aminotransferase (*ALT*) and aspartate aminotransferase (*AST*) are essential aminotransferases in amino acid metabolism and are used to evaluate changes in environmental factors, feeding, growth and development, and liver function test indicators [15]. During the cycle, *ALT* mainly catalyzes the reaction of α-ketoglutarate and aspartate to produce glutamate and oxaloacetate, and *AST* mainly catalyzes the reaction of α-ketoglutarate and alanine to produce glutamate and pyruvate. Glutamate oxaloacetate transaminases (*GOT*) in fish are key enzymes in amino acid metabolism, and their activity reflects not only the extent of amino acid metabolism but also the nutritional status of fish [16]. Under acute temperature stress, the expression levels of *GOT1* and *AST* were altered in the stonefish serum [17]. In mammals, the conversion of α-aminoadipic acid to α-ketoadipic acid by α-aminoadipate transaminase (*AADAT*) is an intermediate step in lysine degradation [18]. All evidence suggests that members of the transaminase gene family play directly or indirectly regulatory roles in responding to

environmental challenges in animals. However, only a few aminotransferase genes have been cloned for functional analysis, and their characteristics are largely unknown.

The Jinjiang oyster *Crassostrea ariakensis* is a foundational organism inhabiting estuaries where the environmental conditions are highly complex, and salinity fluctuations are frequent. To better understand the oyster's tolerance to salinity stress, we conducted a comprehensive analysis of the aminotransferase class I and II gene family in *C. ariakensis* using genomic data. This analysis included studying gene structures, motif compositions, chromosome localization, and phylogenetic relationships. Furthermore, we also utilized real-time fluorescence quantitative PCR (qRT-PCR) technology to investigate the response characteristics of all members of the aminotransferase class I and II gene family in *C. ariakensis* under acute high-salinity stress. The objective was to provide reference data that could facilitate a further exploration of the role of this gene family in the osmotic pressure regulation of *C. ariakensis*.

## 2. Materials and Methods

### 2.1. Identification of Aminotransferase Class I and II Gene Sequences

To identify the members of the aminotransferase class I and II gene family, genome data of *C. ariakensis* (CNA0022698) [19] were obtained from the China National Gene Bank (https://db.cngb.org (accessed on 14 December 2021)) and *Acanthopleura granulata* genome data from the Dryad database (https://datadryad.org/stash/dataset/doi:10.5061/dryad.wstqjq2k9 (accessed on 14 December 2021)). In addition, genome data of *C. virginica* (CNA0022698), *C. gigas* (GCA_902806645.1), *Octopus sinensis* (GCF_006345805.1), and *C. hongkongensis* (GCA_015776775-1) were derived from websites. According to the conserved condition of the *Aminotran_1_2* (PF00155) domain in aminotransferase class I and II family members, the *Aminotran_1_2* Hidden Markov Model (HMM) was downloaded from the Pfam (http://pfam-legacy.xfam.org/ (accessed on 14 December 2021)) database. The complete aminotransferase class I and II family members in the *C. ariakensis* protein data file was acquired using Hmmer (3.0). Then Tbtools (v1.0987663) [20] was used to screen out the *C. ariakensis* aminotransferase class I and II gene family members by comparing the obtained results with the annotation file results.

### 2.2. Bioinformatics Analysis of Aminotransferase Class I and II Gene Sequences in C. ariakensis

An open reading frame (ORF) finder (http://www.ncbi.nlm.nih.gov/gorf/gorf.html (accessed on 25 December 2021)) was used to predict amino acid sequences. The molecular weight (MW) and isoelectric point (pI) of aminotransferase class I and II proteins were identified using the ExPASy Prot-Param tool (https://web.expasy.org/program/ (accessed on 25 December 2021)). The subcellular localization of all identified aminotransferase class I and II proteins was predicted using CELLO v.2.5 (http://cello.life.nctu.edu.tw/ (accessed on 26 December 2021)) and Euk-mPLoc 2.0 (http://www.csbio.sjtu.edu.cn/bioinf/euk-multi-2/ (accessed on 27 December 2021)). The complete aminotransferase class I and II gene families were aligned using ClustalX with default parameters [21]. The phylogenetic tree was constructed via the Neighbor-joining (NJ) algorithm using the MAGE (v11) program. And the gene family arrangement of aminotransferase class I and II in *C. ariakensis* was visualized in the Jalview (v 2.11.2.1) software. The chromosomal distribution of gene families was analyzed to detect the duplication of aminotransferase class I and II gene families using the Tbtools software based on annotation information.

### 2.3. Multiple Comparisons and Structural Analysis of Aminotransferase Class I and II Gene Sequences

To confirm the presence of conserved domains, all aminotransferase class I and II proteins were conducted in *C. ariakensis* using the NCBI Conserved Domain Database (https://www.ncbi.nlm.nih.gov/Structure/cdd/wrpsb.cgi (accessed on 28 December 2021)) and the architectural research tool Tbtools. This allowed us to identify and study the conserved domains within these proteins and their simple modularization. DNA and cDNA sequences

corresponding to each predicted aminotransferase class I and II gene from the *C. ariakensis* genome and transcriptome numbers were used to determine exon size and the location of exon-intron boundaries. The Gene Structure Display Server (GSDS, http://gsds.cbi.pku.edu.cn/ (accessed on 28 December 2021)) was used to generate the exon-intron structure diagram of aminotransferase class I and II genes. Conserved motifs of aminotransferase class I and II proteins were predicted using the Motif Elimination (MEME) software (http://meme-suite.org/tools/meme (accessed on 29 December 2021)).

### 2.4. Salinity Challenge and Sample Preparation

Oysters with an average shell length of 56.85 ± 1.94 mm were harvested in January 2022 from Dingziwan Bay, Shandong Province. Thirty oysters were sampled, equally divided into three groups of ten oysters each, and placed in a 50 cm × 40 cm × 28 cm tank for temporary rearing. Before the stress experiment, the oysters were aerated in seawater at 20 °C and a salinity of 25 for five days. The water was changed once a day and the oysters were fed with *Chlorella vulgaris* in the morning and the evening, respectively. After acclimatization, the oysters were transferred to water with a salinity of 40 for the salinity stress; the other culture conditions were the same as those in the acclimatization stage. Gill tissues were dissected at 0 h, 6 h and 12 h post stress, rapidly frozen in liquid nitrogen, and then stored at −80 °C for RNA extraction.

### 2.5. Expression Analysis of Aminotransferase Class I and II under High-Salt Stress

RNA was extracted using Trizol Reagent (Wuhan Yipu Biotechnology Co., Ltd. Wuhan, China) following the manufacturer's instructions, and the purity and quality of RNA were detected using a microspectrophotometer (NanoPhotometer™, Munich, Germany) and agarose gel electrophoresis. The cDNA was synthesized using HiScript III RT SurperMix for qPCR (Vazyme, Nanjing, China), primers were designed using Primer 6 (Table 1), and qRT-PCR was performed to detect the expressions of aminotransferase class I and II gene members before and after acute salinity stress using β-actin gene as an internal reference. For primer validation, the amplification efficiency should fall within the range of 95% to 100%, and for genes that exceed or fall below this interval, primer redesign is recommended (Table 1). The 20 μL qRT-PCR reaction system contained 0.4 μL of each upstream and downstream primers, 2 μL of template cDNA (500 ng/μL), 10 μL 2 × ChamQ SYBR Color qPCR Master Mix, and 7.2 μL of DEPC water. The reaction procedure consisted of a pre-denaturation at 95 °C for 10 min, 40 cycles of denaturation at 95 °C for 10 s, and annealing at 60 °C for 30 s, followed by melt curves analysis at 95 °C for 15 s, 60 °C for 60 s, and 95 °C for 15 s. After the reaction, the relative gene expression was calculated using the $2^{-\triangle\triangle CT}$ method. One-way ANOVA was performed using SPSS (v26), and the GraphPad Prism (v8.0.2) was used to produce gene expression bar graphs. Three biological replicates and three technical replicates were used for each group.

**Table 1.** Primers used in qRT-PCR experiments.

| Gene Name | Abbreviation | Sequence (5′–3′) | Product Size (bp) | Efficiency (%) |
|---|---|---|---|---|
| Serine palmitoyltransferase 1 | *CarSPTLC1* | GTGCTATCCCAGCTTACTCGAAACG GCTCCAGATCCTCCATGTCATTGTG | 172 | 95.99 |
| Serine palmitoyltransferase 2 | *CarSPTLC2* | TGGTGCTGTTGGAGGTTACATTGG GGAGTAGATAGCCGAGTGGGAGTAC | 140 | 95.09 |
| Serine—pyruvate aminotransferase | *CarSPTa* | GAGGTGCTGATGGATGAC GCTATGGCGTAGGAGAAC | 245 | 96.38 |
| Serine—pyruvate aminotransferase | *CarSPTb* | AACAAGGTGCTGAATCGTA GCGTAATCGGACACATCT | 335 | 97.99 |
| 5-aminolevulinate synthase | *CarALAS* | GGCTTCATCTTCACCACCAGTCTC CTATCAGGTTCTGTCGCAGGTAACG | 244 | 97.21 |

**Table 1.** *Cont.*

| Gene Name | Abbreviation | Sequence (5′–3′) | Product Size (bp) | Efficiency (%) |
|---|---|---|---|---|
| Alanine aminotransferase 2 | *CarALAT2a* | CTCCCGATGCTTTCTACTGCTTCC<br>CCTCCTTCTCTCCGAACCCACTC | 213 | 96.23 |
| Alanine aminotransferase 2 | *CarALAT2b* | TGCCTTCCCTCAAATCAAGATGCC<br>GTTCCTGGTTGCTGTCCGAATCC | 397 | 98.11 |
| Aspartate aminotransferase | *CarASTa* | TGGTTGGGAGAAGTGAAAGGAATGG<br>GGTCAATGATGTGTTGCCAGTTACG | 165 | 97.20 |
| Aspartate aminotransferase | *CarASTb* | GTCATAACCCAACTGGAGTGAACCC<br>GTTACCAGAAGCGAAGCCCTGATAG | 185 | 95.01 |
| Aspartate aminotransferase, cytoplasmic | *CarASTc* | GCTCTAGGATTCTGGATGCTGTTGG<br>TGTCTGGATTCCTCGCCTCTTGAG | 121 | 98.86 |
| Kynurenine—oxoglutarate transaminase 3 | *CarKAT3a* | TTCGCCCCACCAAAGCACATTC<br>GCTGCTGAAGTTCTCGGCTGTAG | 89 | 99.24 |
| Kynurenine—oxoglutarate transaminase 3 | *CarKAT3b* | GCTCCAGATGATGGCAGACTTGTG<br>AACATCCCAGGCAGAGAGGCTATC | 118 | 97.35 |
| Kynurenine/alpha-aminoadipate aminotransferase | *CarAADAT* | TCAAAGCCGTATGTTCCCAGTCTTC<br>CAGTCACAAACCCCACCCTCATTC | 312 | 97.68 |
| Tyrosine aminotransferase | *CarTAT* | CAACGGCTACAACCCATCCATAGG<br>CAGCCACCAGTAAGAACCACATCC | 164 | 95.29 |
| Alanine—glyoxylate aminotransferase 2 | *CarAGT2* | AATGACTTGGCGGTTCTG<br>CTGTGTTGTTCCTCCTACTC | 381 | 97.82 |
| Serine/threonine-protein kinase 31 | *CarSTK31* | GCGAGCCGTCAGGAATTAGGTTATC<br>GCTGGCAGGTTCATAGAGTTGGTAG | 169 | 98.00 |
| 2-amino-3-ketobutyrate coenzyme A ligase | *CarKBL* | CCAGTTGTTGCTTGTGCCAGTAAAG<br>GCGGAATCGTTTGGTGTTGTTGAG | 135 | 98.90 |
| Uncharacterized protein | *CarYER152C* | TGTGATCTGTTGTTTGGTGGACCTG<br>AGAAGAGCCTGCGTTGCCTTTAAG | 402 | 99.68 |
| | *β-actin* | CTGTGCTACGTTGCCCTGGACTT<br>TGGGCACCTGAATCGCTCGTT | 122 | 94.77 |

## 3. Results

### 3.1. Gene Identification and Protein Characteristic Analysis

A total of 18 aminotransferase class I and II gene families were identified in the *C. ariakensis* genome, namely, *SPTLCa* and *SPTLCb*, *SPTa* and *SPTb*, *ALAT2a* and *ALAT2b*, *KAT3a* and *KAT3b*, *ASTa*, *ASTb* and *ASTc*, as well as *ALAS*, *AADAT*, *ATA*, *AGT2*, *STK31*, *KBL*, and *YER152C*. The gene names, protein sequence lengths, and isoelectric points of each family member are shown in Table 2. The aminotransferase class I and II gene families had amino acid numbers ranging from 407 to 1982 and molecular weights ranging from 45.71 kDa to 221.36 kDa, with *CarSPTLC1* being the longest at 1982 aa and *CarASTb* being the shortest at 407 aa. The protein isoelectric points ranged from 5.49 to 9.03. The isoelectric point of *CarSTK31* was the smallest and that of *CarALAT2b* was the largest. The protein instability point coefficients of most family members were less than 40, including *CarSPTLC2*, *CarSPTb*, *CarALAT2a*, *CarASTa*, *CarASTb*, *CarKAT3a*, *CarKAT3b*, *CarAADAT*, *CarAGT2*, *CarSTK31*, and *CarKBL*, while the others were greater than 40. The results of the protein hydrophobicity calculations showed that aminotransferase classes I and II had different degrees of hydrophilicity. The subcellular localization results showed that aminotransferase class I and II proteins were generally localized in the cytoplasmic and mitochondrial compartments. Notably, *CarSPTb*, *CarKAT3a*, *CarAGT2*, and *CarKBL* were designated only in the mitochondrial compartment, while *CarALAT2a*, *CarASTb*, *CarKAT3b*, and *CarTAT* were localized only in the cytoplasmic compartment.

### 3.2. Chromosome Localization Analysis and Gene Covariance

The positions of aminotransferase class I and II gene family members on the *C. ariakensis* chromosome are shown in Figure 1. *C. ariakensis* has 2n = 20 chromosomes, and 18 genes of the aminotransferase class I and II family were localized on eight chromosomes. No aminotransferase class I and II genes were found on chromosome 1 and chromosome 10. *CarSPTLC1* was localized on chromosome 2, while *CarALAT2a* and *CarALAS* on chromosome 3, *CarASTa*, *CarASTb*, *CarALAT2b*, *CarAGT2*, and *CarTAT* on chromosome 4, *CarKAT3b*

on chromosome 5, *CarSPTLC2* and *CarASTa* on chromosome 6, *CarAADAT* and *CarSTK31* on chromosome 7, *CarKBL*, *CarASTc*, and *CarKAT3a* on chromosome 8, and *CarYER152C* and *CarASTb* on chromosome 9.

**Table 2.** Protein composition and physicochemical properties of the Aminotransferase class I and II gene family in *C. ariakensis*.

| Gene ID | Gene Name | Protein Length (aa) | Molecular Weight (kDa) | pI | Instability Index | Grand Average of Hydropathicity | Localization |
|---|---|---|---|---|---|---|---|
| Ori.tig00000232.98 | *CarSPTLC1* | 1982 | 221.36 | 5.91 | 40.44 | −0.429 | NC [a], CP [b] |
| Ori.tig01225668.108 | *CarSPTLC2* | 1107 | 123.53 | 8.82 | 34.46 | −0.215 | NC [a], CP [a], MB [b] |
| Ori.tig00000276.148 | *CarSPTa* | 415 | 45.95 | 8.49 | 40.33 | −0.05 | Mit [a,b], CP [a] |
| Ori.tig00000013.86 | *CarSPTb* | 431 | 47.13 | 8.82 | 35.18 | −0.144 | Mit [a,b] |
| Ori.tig00014637.33 | *CarALAS* | 615 | 57.73 | 7.12 | 44.74 | −0.295 | Mit [a,b], CP [a] |
| Ori.tig01225679.19 | *CarALAT2a* | 529 | 58.74 | 7.02 | 32.77 | −0.208 | CP [a,b] |
| Ori.tig00057033.282 | *CarALAT2b* | 825 | 92.59 | 9.03 | 41.08 | −0.35 | Mit [a,b], CP [a] |
| Ori.tig00001151.290 | *CarASTa* | 428 | 47.04 | 8.95 | 32.22 | −0.25 | Mit [a,b], CP [a] |
| Ori.tig01227070.450 | *CarASTb* | 407 | 45.71 | 7.13 | 30.46 | −0.237 | CP [a,b] |
| Ori.tig00000007.252 | *CarASTc* | 550 | 62.22 | 5.80 | 41.40 | −0.095 | PM [a], CP [b] |
| Ori.tig00000007.312 | *CarKAT3a* | 468 | 52.51 | 8.61 | 39.30 | −0.31 | Mit [a,b] |
| Ori.tig00057021.276 | *CarKAT3b* | 680 | 76.18 | 5.57 | 35.95 | −0.146 | CP [a,b] |
| Ori.tig00000052.211 | *CarAADAT* | 459 | 51.1 | 5.87 | 39.9 | −0.218 | NC [a], CP [a,b] |
| Ori.tig00057030.149 | *CarTAT* | 434 | 48.51 | 7.22 | 44.37 | −0.249 | CP [a,b] |
| Ori.tig00057033.171 | *CarAGT2* | 488 | 53.73 | 8.56 | 29.36 | −0.097 | Mit [a,b] |
| Ori.tig00000599.319 | *CarSTK31* | 478 | 53.52 | 5.49 | 36.65 | −0.093 | CP [a], IM [b] |
| Ori.tig01225420.248 | *CarKBL* | 425 | 46.34 | 8.68 | 29.02 | −0.167 | Mit [a,b] |
| Ori.tig00002832.215 | *CarYER152C* | 461 | 52.18 | 6.68 | 48.08 | −0.268 | NC [a], CP [a,b] |

Notes: Cytoplasmic (CP); Nuclear (NC); InnerMembrane (IM); Mitochondrial (Mit); PlasmaMembrane (PM). The result was predicted by [a]: Euk-mPLoc 2.0; [b]: CELLO v.2.5.

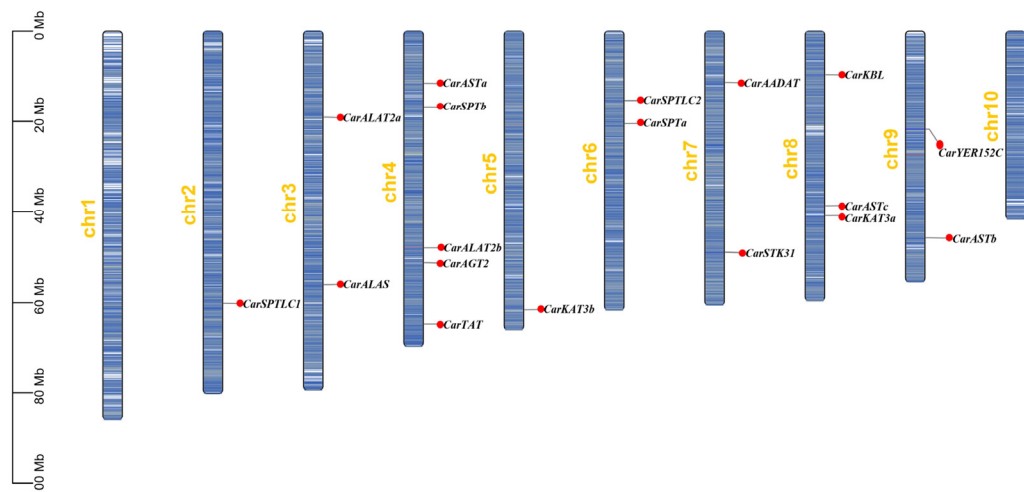

**Figure 1.** Chromosome location diagram of *C. ariakensis* aminotransferase class I and II gene family.

To further study the phylogenetic relationship of the common oysters in northern China (*C. ariakensis* and *C. gigas*), a collinearity analysis diagram was created (Figure 2). The location distribution of aminotransferase class I and II genes on the chromosomes of the two oysters was different. For example, *CarSPTLC1* on chromosome 2 of *C. ariakensis* was located on chromosome 5 of *C. gigas*, and *CarASTa*, *CarASTb*, *CarALAT2b*, *CarAGT2*, and *CarTAT* on chromosome 4 of *C. ariakensis* were located on chromosome 5 of *C. gigas*. *CarKAT3b* on chromosome 5 of *C. ariakensis* was positioned on *C. gigas* chromosome 7, while *CarSPTLC2* and *CarASTa* on chromosome 6 of *C. ariakensis* were located on *C. gigas* chromosome 3. *CarAADAT* and *CarSTK31* on chromosome 7 of *C. ariakensis* were arranged on *C. gigas* chromosome 10, while *CarKBL*, *CarASTc*, and *CarKAT3a* on chromosome 8 of *C. ariakensis* were positioned on *C. gigas* chromosome 1. Finally, *CarYER152C* and *CarASTb*

on chromosome 9 of *C. ariakensis* were located on *C. gigas* chromosome 4. It was important to note that CarALAT2a on *C. ariakensis* chromosome 3 was located on *C. gigas* chromosome 8. However, the *CarALAS* gene was not identified in *C. gigas*.

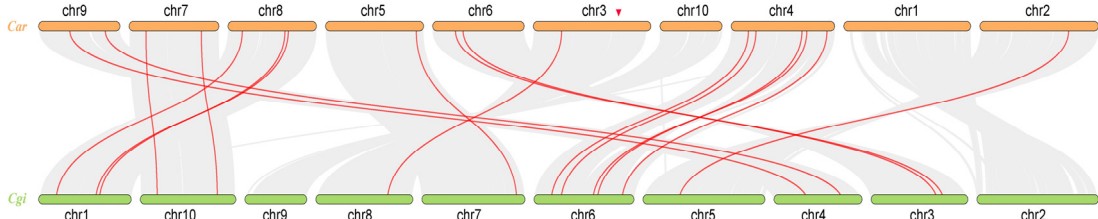

**Figure 2.** Covariance analysis of aminotransferase class I and II genes in *C. ariakensis* and *C. gigas*. The red triangle represents *CarALAS*. Car means Jinjiang oyster (*C. ariakensis*); Cgi means Pacific oyster (*C. gigas*).

### 3.3. Gene Structural Characterization

The structure of the 18 members of the aminotransferase class I and II gene family in *C. ariakensis* is illustrated in Figure 3a. The total length of different genes on the chromosome varied, with *CarYER152C* being the longest and *CarSPTLC1* being the shortest. Moreover, the number and length of introns were different among the genes; the *CarYER152C* gene had the longest intron.

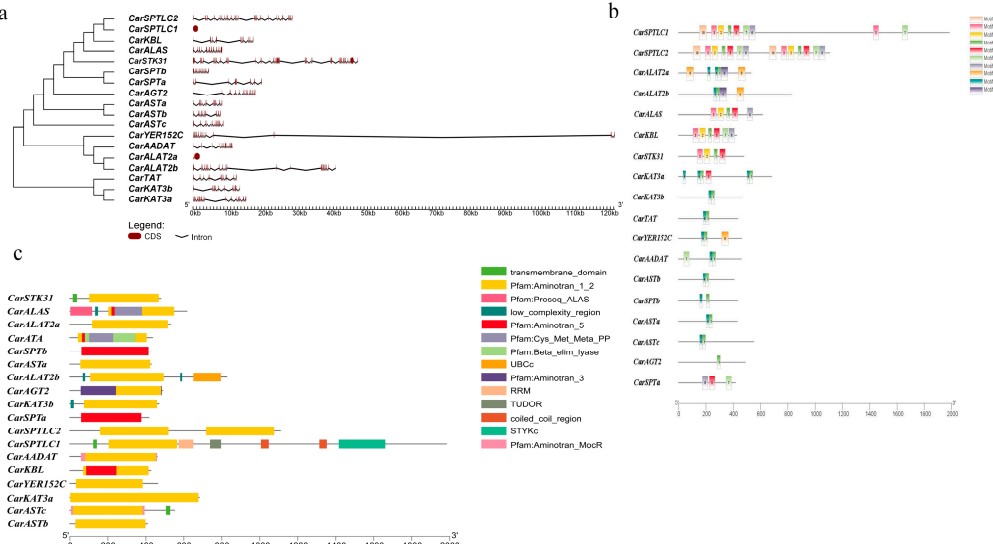

**Figure 3.** Schematic diagram of *C. ariakensis* gene structure. (**a**) Schematic representation of the exon and CDS structure of the gene. The red box indicates CDS and the black line indicates intron. (**b**) Motif analysis of different genes, different colors indicate different motifs. (**c**) Analysis of the structural domains of different genes, with different colors indicating different structural domains.

Conserved motif analysis of the aminotransferase class I and II gene family in the *C. ariakensis* revealed the presence of 10 motifs in all members (Figure 3b). However, the number of motifs varied among the 18 family members to the extent that *CarAGT2* only had motif 1, while *CarSPTLC2* had 14 motifs. Most family members possessed motifs 1, 2, and 4, of which motifs 1 and 4 were the most frequent.

Conserved structural domain analysis indicated that all aminotransferase class I and II proteins possessed the *Aminotran_1_2* structural domain, which catalyzes the reaction between amino acids and α-keto acids (Figure 3C). Notably, *CarSPTa* and *CarSPTb* only had the *Aminotran_5* domain (PF00266).

### 3.4. Evolutionary Analysis of Aminotransferase Class I and II Gene Family Members

The clustering results are shown in Figure 4. The gene family members of *C. ariakensis*, *C. gigas*, *C. hongkongensis*, *C. virginica*, and *O. sinensis* clustered closer together, while *A. granulate* (the West Indian fuzzy chiton) clustered more distantly. Within the four shellfish species, most members from the same subfamily first clustered together and then clustered with members of other subfamilies. Most members of the aminotransferase class I and II gene family in *C. ariakensis* clustered closer to *C. gigas* and *C. hongkongensis*, followed by *C. virginica*, while *A. granulata* was more distant, which is essentially the same as the traditional classification status.

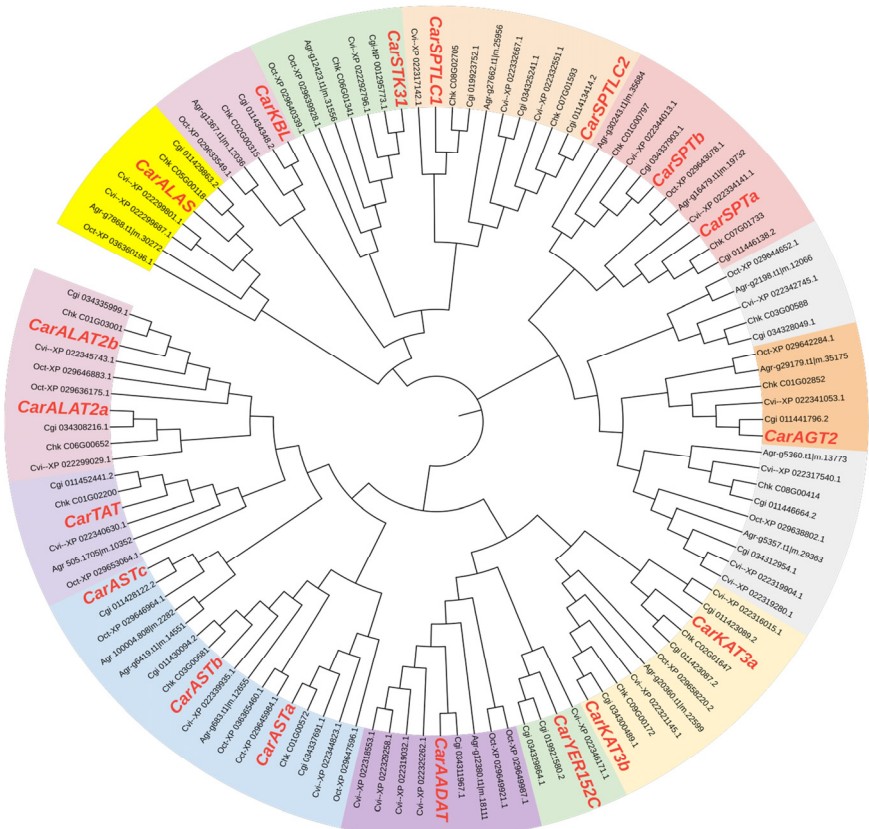

**Figure 4.** Phylogenetic tree of aminotransferase class I and II gene family members among six species. Abbreviations: Car: *Crassostrea ariakensis*; Cgi: *Crassostrea gigas*; CH: *Crassostrea hongkongensis*; Cvi: *Crassostrea virginica*; Osi: *Octopus sinensis*; Agr: *Acanthopleura granulate*.

### 3.5. Analysis of Gene Expression Patterns under Acute High-Salt Stress

The expression levels of most genes of the aminotransferase class I and II gene family changed significantly at 6 h and 12 h when compared to those at 0 h. As shown in Figure 5, the expression trends of family members were similar, showing an overall trend of increasing and then decreasing. The expression changes of *CarALAT2a*, *CarAGT2*, and *CarKAT3b* were noteworthy. The expression trend of *CarALAT2a* initially increased and then decreased, there were significant differences at 6 h and 12 h compared to the relative expression at 0 h ($p < 0.01$), and the highest relative expression at 6 h was significantly different from that at 12 h ($p < 0.05$). The expression trend of *CarAGT2* increased with time, and the expression reached the highest level at 12 h, which was 0.5 times higher than that at 0 h ($p < 0.01$). The expression pattern of *CarKAT3b* was the same as that of *CarALAT2a*, and the relative expression was 1.0 times at 6 h and 0.5 times at 12 h higher than that at 0 h, respectively.

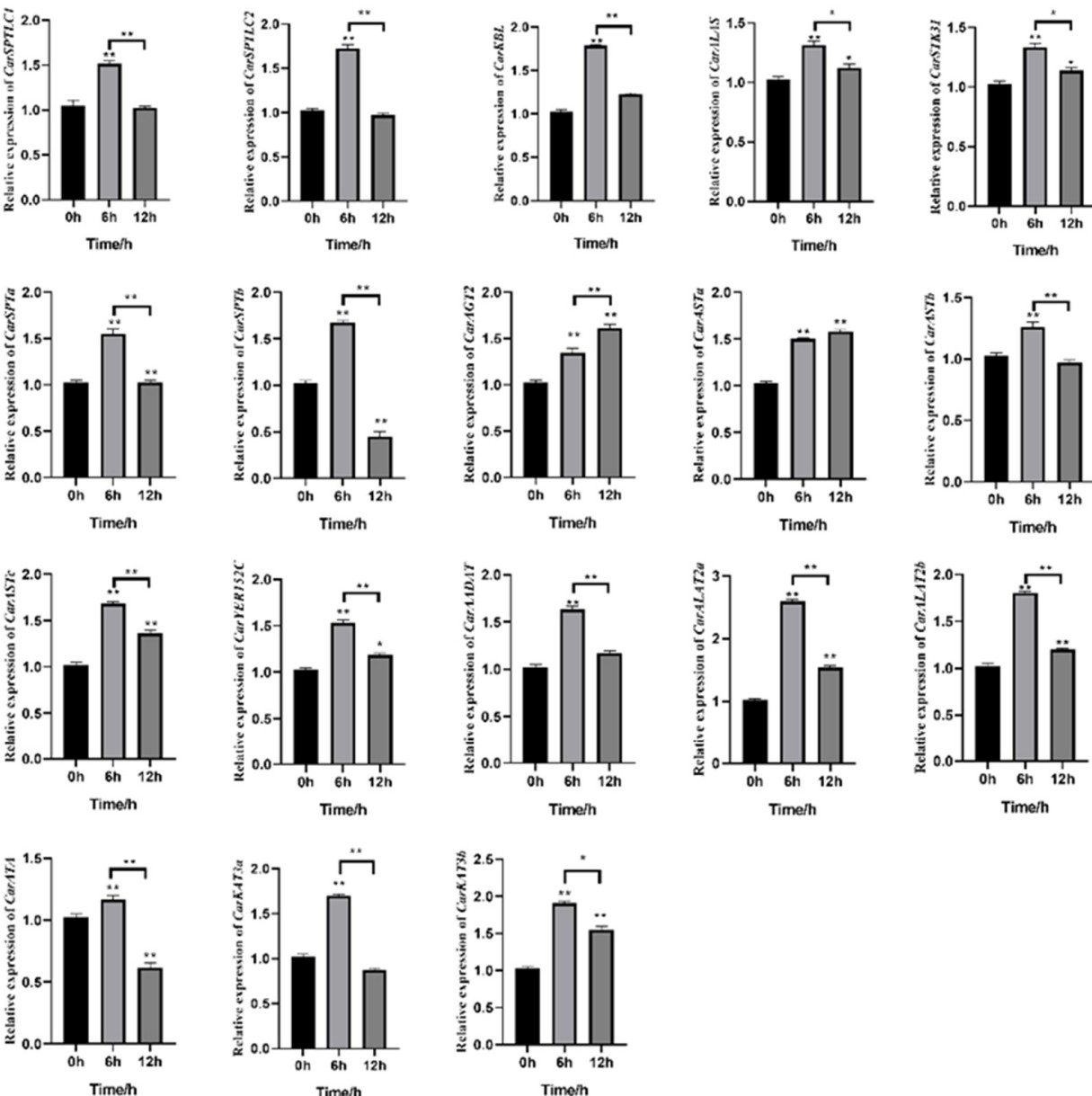

**Figure 5.** Relative expression levels of aminotransferase class I and II gene family members in the gills of *C. ariakensis*. Asterisks indicate significant differences compared to the 0 h time point as follows: * indicated $p < 0.05$ or ** indicated $p < 0.001$.

## 4. Discussion

It has been reported that aminotransferase classes I and II include aspartate aminotransferases, aromatic amino acid aminotransferases, alanine aminotransferases, and histidine phosphate aminotransferases [13]. In this study, a total of 18 aminotransferase class I and II gene family members were identified in the *C. ariakensis* genome, including *SPTLCa* and *SPTLCb*, *SPTa* and *SPTb*, *ALAT2a* and *ALAT2b*, *KAT3a* and *KAT3b*, *ASTa*, *ASTb* and *ASTc*, as well as *ALAS*, *AADAT*, *ATA*, *AGT2*, *STK31*, *KBL*, and *YER152C*. The identified genes are of significant importance for *C. ariakensis*, as they are involved in several essential biological processes. Firstly, they participate in protein synthesis, a fundamental mechanism responsible for generating the necessary proteins required for various cellular functions, growth, and development [22]. Secondly, these genes are key players in amino acid metabolism, facilitating processes such as the breakdown, synthesis, and interconversion of amino acids. This metabolic pathway ensures a proper balance of amino acids, vital for meeting the

*C. ariakensis*'s physiological needs [23]. Furthermore, the genes are essential for energy production within *C. ariakensis*. They actively engage in metabolic pathways like the tricarboxylic acid cycle (TCA cycle) or Krebs cycle, contributing to the generation of energy (ATP), which is crucial for the organism's survival and various activities [24]. These genes are responsible for the synthesis of important amino acids, specifically leucine and tryptophan, of which leucine as a branched-chain amino acid plays a critical role in protein synthesis and muscle development. On the other hand, tryptophan serves as a precursor for various bioactive compounds and neurotransmitters [25].

In the present study, the subcellular localization results showed that most aminotransferases class I and II gene members were localized in mitochondria and cytoplasm, which is consistent with the fact that amino acid metabolism occurs in mitochondria [26]. In addition, the aminotransferases class I and II exhibited different degrees of hydrophilicity; however, hydrophilic amino acids can form intermolecular hydrogen bonding networks to enhance the stability of proteins and keep the activity of proteins and enzymes at a certain level, thus playing a role in regulating the structure of proteins [27]. Since the pI is mainly determined by the number of acidic and basic amino acids [28], and most *C. ariakensis* aminotransferases class I and II member proteins have a pI less than 7.0, they may be a class of acidic proteins.

Motifs such as 1, 4, and 7, were shared by some members of the *C. ariakensis* aminotransferase class I and II gene family, indicating that these gene families are highly conserved. However, some motifs were only present in some subfamilies, suggesting different subfamilies with functional diversity. For example, motif 1 was the only one present in *CarAGT2*, Aminotransferases and other enzymes display some structural domains that are highly conserved throughout evolution. *Aminotran_5* (PF00266), which encodes a protein that is a phosphoserine aminotransferase, regulates immune function and is involved in oxidative stress [29] including *CarSPTa*, *CarSPTb*, and *CarKBL*. Transmembrane domains may function as membrane receptors, membrane-anchored proteins, or ion channel proteins localized on the membrane [30,31]. These transmembrane structural domains are usually hydrophobic, while the extramembrane structures are generally hydrophilic [32]. This is consistent with the hydrophilicity of those genes in this study that are localized on the mitochondrial and plasma membrane. Li et al. [33] found that *C. ariakensis* has high gene covariation with the oyster *C. hongkongensis*. The *C. ariakensis* aminotransferases class I and II gene family first clustered with *C. hongkongensis* and *C. gigas*, and later with other species such as *C. virginica*. Additionally, evolutionary analysis on six species revealed that members of the same subfamily from different species clustered together first, except for *A. granulata*, which is generally consistent with traditional taxonomic relationships.

Studies have also shown that aminotransferases classes I and II are involved in regulating osmotic pressure and protein synthesis and degradation [34]. Specifically, these enzymes can control the rate of protein synthesis and degradation by regulating the metabolic pathways of amino acids to maintain stable intracellular osmolarity [35]. Aminotransferases classes I and II have been cloned and expressed in *Litopenaeus vannamei*, which showed their important role in osmolarity regulation [36]. In *Anguilla anguilla*, a rapid increase in aminotransferase activity was observed in the kidney when responding to the changes in environmental salinity [37]. In *Lutjanus sanguineus*, the expression of alanine aminotransferase 2 genes increased with time under acute salinity stress in a highly osmotic environment [38]. In this study, the aminotransferase class I and II gene family participated in the response to acute high-salinity stress in *C. ariakensis*. However, different members from this gene family exhibited varying expression patterns during the stress response. Most members showed an initial increase in expression followed by a subsequent decrease, and these differences were statistically significant. Considering the results of the bioinformatics analysis of the aminotransferase class I and II gene family members, it was believed that this gene family plays a crucial role in the osmotic regulation process of *C. ariakensis*. These findings can provide valuable reference data for further investigating the salt adaptation mechanisms of *C. ariakensis*.



## 5. Conclusions

In conclusion, we have identified the aminotransferase class I and II gene family in *C. ariakensis* using bioinformatics methods and validated their expression characteristics under acute high-salt stress. A total of 18 aminotransferase class I and II gene members were identified, among which 11 members showed stable physicochemical properties. These members are distributed on eight chromosomes and are localized in either the cytoplasm or mitochondria. All members possess the *Aminotran_1_2* domain (PF00155), which regulates osmotic pressure by facilitating the conversion of amino acids and ketones. Under acute high-salt stress, the expression levels of most genes initially increased and then decreased. However, as the stress duration extended, the expression of the *CarAGT2* and *CarKAT3b* genes showed an upward trend. These results provide valuable information on the aminotransferase class I and II gene family in bivalves and lay the groundwork for further exploration of this family's role in osmotic stress regulation.

**Author Contributions:** Y.Z.: conceptualization, supervision, writing—review and editing, and funding acquisition; G.G.: formal analysis, visualization, conceptualization, software, data curation, and writing—original draft; Z.L. (Zhihong Liu) and L.Z.: validation, resources, and writing—review and editing; Z.L. (Zhuanzhuan Li) and X.S.: resources and visualization; T.Y.: resources and supervision. All authors have read and agreed to the published version of the manuscript.

**Funding:** This research was funded by the National Key R&D Program of China, 2022YFD2400100; the Central Public-interest Scientific Institution Basal Research Fund, CAFS, 2021XT0101; the Agroecological Environment Survey and Monitoring Services, A060102; the Laoshan Laboratory (LSKJ202203803) and the Project of Yellow River Fisheries Resources and Environment Investigation from the MARA, P. R. China.

**Institutional Review Board Statement:** The study was approved by the Laboratory Animal Care, Ethics and Safety Review Form of the Yellow Sea Fisheries Research Institute, Chinese Academy of Fishery Sciences (Approval No. YSFRI-2023035).

**Data Availability Statement:** The data used throughout the text were downloaded from publicly available databases. The data were downloaded from the National Center for Biotechnology Information (NCBI), we downloaded the following GenBanks: CNA0022698, GCA_902806645.1, GCF_006345805.1, and GCA_015776775-1 (https://www.ncbi.nlm.nih.gov/datasets/genome/) (accessed on 14 December 2021). And downloaded from the China National Gene Bank: CNA0022698 (https://db.cngb.org) (accessed on 14 December 2021).

**Acknowledgments:** We are thankful to Researcher Biao Wu of the Shellfish Breeding Laboratory, Yellow Sea Fisheries Research Institute, Chinese Academy of Fishery Sciences, for their technical assistance and expertise in various aspects of this research. We would also like to extend our appreciation to Xi Chen from the Ocean University of China for their guidance on the analytical techniques used in this study.

**Conflicts of Interest:** The authors declare no conflict of interest.

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
