# Peer review of "Aminotransferase Class I and II Gene Family in the Jinjiang Oyster (Crassostrea ariakensis): Genomewide Identification, Phylogenetic Analysis and Expression Profiles after Salinity Stress"

_fishes, doi:10.3390/fishes8090459_

Round 1

Reviewer 1 Report

The manuscript by Ge et al., describes a study designed to identify members of the Aminotransferase class I and II gene families in the gills of Jinjiang oyster (C. ariakensis), characterize their gene structure, chromosomal location and phylogeny, and investigate the mRNA expression responses (6 and 12 h) to an acute salinity challenge elicited by direct transfer from 25 to 40 ppt. The main goal was to provide baseline reference data for future research into the osmoregulatory mechanisms of C. ariakensis.

General Comments:

Overall, this is an interesting study that should advance our knowledge regarding the role of aminotransferases in animal osmoregulation.  The manuscript is generally well organized, but some of the grammar and spelling could be improved.  I only have some minor concerns for the authors to consider below.

Specific Comments:

Line 20:  here and throughout the manuscript, please check the spelling of the species name, ariakensis

Line 43:  here and throughout, I suggest the authors take care when using the term “adapt” as adaptation refers specifically to changes in gene frequencies in populations over generations.  I think in many cases “acclimatize” would be the more appropriate term. 

Line 59:  “In living organisms” is unnecessary and can be omitted

Line 76:  If “their activity reflects not only the extent of amino acid metabolism” this begs the question, what else does their activity reflect?

Line 159:  The 2 delta delta approach assumes that reactions efficiencies are near 100% or at the very least similar in efficiency to the normalizing gene.  Did the authors confirm this?  Were the amplicons sequence verified?

Table 1:  I suggest adding reaction efficiencies and amplicon sizes to this table

Section 2.4:  What were the volumes of the exposures, the density of oysters in each exposure vessel, and the level of exposure replication?

Figure 5:  suggest revising first sentence of caption to:  “Relative expression levels of Aminotransferase class I and II gene family members in gills of C. ariakensis.” and suggest revising second sentence of caption to:  “Asterisks indicate significant differences compared with 0 h time point as follows:  *P < 0.05 or **P < 0.001.”

Line 275:  please change “significantly” to “significant”

Lines 293-294:  I do not follow the logic in this sentence, please clarify

Line 301:  I think the authors mean “with” instead of “were”

Lines 307-309:  this sentence is awkwardly structured; please revise

Line 310-311:  what do the author’s base this statement on? the justification is not presented well in the context of this section; this is discussed in more detail in the following paragraph, so perhaps this should be omitted here.

There are a number of grammatical and spelling errors in the manuscript.  I have pointed out some of these, but suggest the authors go through and carefully review again before resubmission.

Author Response

Dear reviewers
Thank you for your hard work on this paper. We have made careful revisions in accordance with your comments and have highlighted them in red.

Reviewer 2 Report

General Comments:

The paper "Aminotransferase class I and II gene family in Jinjiang oyster (Crassostrea ariakensis): genomewide identification, phylogenetic analysis and expression profiles after salinity stress" presents an investigation into the Aminotransferase class I and II gene family in Jinjiang oyster, focusing on genomic characteristics, chromosomal localization, phylogenetic analysis, and expression profiles under salinity stress. The study contributes valuable insights into the osmotic pressure regulation mechanisms in this mollusk. The paper is well-structured and covers essential aspects of the research. The language is generally clear, but there are some areas that require improvement for clarity and precision. The study's significance, methodology, and findings are well-conveyed. Below are major and minor points for consideration.

Major Points:

1.     Abstract Clarification: The abstract provides a succinct overview of the study, but it could be improved by clarifying the significance of the findings in the context of existing knowledge. For instance, mentioning that the identification of these genes and their expression patterns under salinity stress adds valuable information to the understanding of osmoregulation mechanisms in bivalve mollusks would enhance the abstract's impact.

2.     Introduction Relevance: The introduction provides ample background on salinity stress and its effects on shellfish. However, it would be beneficial to clarify early on the importance of aminotransferases in this context, especially in relation to osmoregulation. Highlighting how these enzymes contribute to amino acid metabolism, which is crucial for osmotic pressure regulation, could better connect the introduction to the research focus.

3.     Gene Naming and Abbreviations: The gene names and abbreviations introduced in the text are numerous and may become confusing for readers. Providing a concise table or list of gene names along with their full forms and abbreviations would aid readers in understanding the text and following the discussion.

Minor Points:

1.     Language Clarity: There are instances where the language could be clearer. For example, in the abstract, the sentence "Most genes expressions increased firstly and then decreased from 0 h to 12 h challenged by acute stress of artificial seawater with salinity of 40." could be rephrased for clarity: "Most gene expressions showed an initial increase followed by a decrease between 0 h and 12 h when challenged by acute stress using artificial seawater with a salinity of 40."

2.     Line 105: Replace ‘National Gene Bank’ with ‘China National GeneBank DataBase’

The language requires editing. There are instances where the language could be clearer. For example, in the abstract, the sentence "Most genes expressions increased firstly and then decreased from 0 h to 12 h challenged by acute stress of artificial seawater with salinity of 40." could be rephrased for clarity: "Most gene expressions showed an initial increase followed by a decrease between 0 h and 12 h when challenged by acute stress using artificial seawater with a salinity of 40."

Author Response

(The authors gave the same response as above.)

Round 2

Reviewer 2 Report

The authors have addressed my comments, and I can see that the paper is improved and suitable for publication.

Acceptable.